# Transformation of the head-direction signal into a spatial code

Adrien Peyrache [1,2], Natalie Schieferstein[1,4] & Gyorgy Buzsáki [1,3]

Animals integrate multiple sensory inputs to successfully navigate in their environments. Head direction (HD), boundary vector, grid and place cells in the entorhinal-hippocampal network form the brain's navigational system that allows to identify the animal's current location, but how the functions of these specialized neuron types are acquired remain to be understood. Here we report that activity of HD neurons is influenced by the ambulatory constraints imposed upon the animal by the boundaries of the explored environment, leading to spurious spatial information. However, in the post-subiculum, the main cortical stage of HD signal processing, HD neurons convey true spatial information in the form of border modulated activity through the integration of additional sensory modalities relative to egocentric position, unlike their driving thalamic inputs. These findings demonstrate how the combination of HD and egocentric information can be transduced into a spatial code.

[1] Neuroscience Institute, School of Medicine, New York University, 450 East 29th Street, New York City, New York 10016, USA. [2] Department of Neurology and Neurosurgery, Montreal Neurological Institute, McGill University, 3801 University Street, Montreal, Quebec, Canada H3A 2B4. [3] Center for Neuroscience, New York University, New York City, New York 10016, USA. [4] Present address: Institute for Theoretical Biology, Department of Biology, Humboldt-Universität zu Berlin, 10115 Berlin, Germany. Correspondence and requests for materials should be addressed to A.P. (email: adrien.peyrache@mcgill.ca) or to G.Bák. (email: gyorgy.buzsaki@nyumc.org)

Sensory signals are processed by the central nervous system in a hierarchical manner[1]. As one moves from the periphery to the core of the brain, the controlling role of sensory inputs on neuronal firing patterns decreases, whereas higher-level features, including multimodal information and accumulated memory, play an increasing role[2]. Striking examples of such transformation are the integration of sensory features into spatial information by the grid cells in the entorhinal cortex[3,4] and place cells in the hippocampus[5,6]. Together, they form the basis of a cognitive map[6] and convey the highest amount of spatial information in the limbic system[7,8]. Yet, it is not well understood how spatial information arises from sensory inputs[9].

The head direction system[10–14] is a critical part of the navigation system[4,15]. The head-direction signal originates in the vestibular system[11–13] and is updated by external sensory inputs, including visual information[16–19] that references the signal to the environment. The HD signal thus constitutes the simplest allocentric signal of the navigation system. It is conveyed to the parahippocampal region by the antero-dorsal nucleus of the thalamus (ADn)[11,13]. The post-subiculum (PoSub) is the main cortical recipient of ADn output[20–22] and acts as a relay of HD signal to the other structures of the parahippocampal system[20,23,24]. In addition, the PoSub receives inputs from different sensory and association cortices[20] and, thus, occupies a central position in the chain of spatial information processing. In turn, the feedback arising from the PoSub onto thalamic HD neurons updates the HD representation and aligns it to the external world[25]. The HD signal, conveyed directly[22] or indirectly[20,23,24] by ADn neurons, is present in all structures of the parahippocampal circuit[14,16,26–30].

Using various empirical metrics and, often, arbitrary threshold criteria, a variety of neuron classes have been segregated and named[29–40], although the boundaries among these classes are not uncontested[41]. Whereas place cells in the hippocampus and grid cells in the entorhinal cortex are undisputed classes of the brain's navigation system, most neurons in the parahippocampal regions represent intermediate forms. Many neurons unite two or more features (referred to as conjunctive cells)[39], for example conjunctive grid cells with both spatial and HD information[29,31,32]. Border cells or boundary vector cells observed within the medial entorhinal cortex and other parahippocampal structures, fire preferentially at the edges of the environment or signal the distance from the animal to the borders[37,38,40]. Border and HD neurons emerge at the earliest stage of ontogenetic development and precede the development of grid cells and place cells[42–46]. HD neurons and border cells, therefore, may be considered as fundamental building blocks of the spatial code[9]. In support of this hypothesis, grid representation is severely impaired after destruction of ADn neurons[15]. Furthermore, the geometry of the environment exerts a strong influence on the development and firing patterns of place and grid fields[9,47,48].

Despite all these important results, it remains to be demonstrated how activity of HD neurons, sensory information and active exploration by the animal interact and assist in the emergence of a spatial map. In the present experiments, we demonstrate how the animal's behavior itself can influence spatial metric[49]. Specifically, we present evidence that, because animal's heading is not homogeneously distributed in space, HD cells in the thalamus (ADn) convey spatial information from the viewpoint of a downstream reader. In contrast, HD cells in the PoSub convey true spatial information by combining the allocentric HD information with body-centered, egocentric signals, such as the relationship between the ambulatory pattern of the mouse and boundaries of the environment. Such conjunction may be the first stage of computation that integrates primary information to establish the cognitive map.

## Results

**HD cells are influenced by ambulatory constraints.** By definition, a 'pure' head direction (HD) signal is affected by the HD of the animal and nothing else. On the other hand, the spatial specificity of spiking of HD neurons is inevitably affected by the presence of environmental boundaries[33,50,51]. Since not all head directions can be displayed equally near the walls, this bias results in a non-uniform distribution of spikes, as illustrated for an example HD neuron from the ADn (Supplementary Fig. 1a). This neuron exhibits strikingly higher firing rates along the walls parallel with the preferred orientation. As a result of such physical constraint, the cell conveys spatial information. However, this spatial information is spurious[33,50], and can be attributed to the wall-constrained non-uniform distribution of spikes. Various solutions have been suggested to separate the independent contribution of place and HD tuning to the firing of a HD neuron[50,51]. However, those methods usually require high spatial sampling. To estimate the neuron's true spatial information, independent of its HD tuning, we used a straightforward approach: we generated series of random spike trains drawn from a Poisson distribution depending exclusively on the current heading of the animal at each point in time and the cell's HD tuning curve. We therefore defined the 'unbiased spatial information' conveyed by a neuron, as the difference between the observed and control (i.e., HD-corrected) information per spike. An example of such a spike train displays a very similar non-uniform spatial distribution (Supplementary Fig. 1a), resulting in equally high level of spatial information. The low unbiased spatial information of ADn HD neurons shows that tthey were only modulated by the head-direction (Supplementary Fig. 1b–d).

Animals inevitably exhibit biased behavior while exploring an open environment with boundaries (Supplementary Fig. 1e, f). As a result, the more HD information a HD neuron conveys, the more spatially modulated it is, a relationship that does not hold when considering unbiased information (Supplementary Fig. 1g). However, this apparent relationship between HD and non-corrected spatial information was preserved when spatial information was evaluated by a more conservative cross-validated procedure (Supplementary Fig. 2), meaning that it did not result from a measure bias and suggesting that a naive downstream reader neuron or neuron group with no knowledge of the HD tuning of these neurons could extract spatial information from the spike trains. In addition, the positive correlation between HD and spatial information was present not only in the open field but even more so in a multi-arm maze where the mouse behavior is even more constrained (Supplementary Fig. 3).

**HD cells in the post-subiculum convey spatial information.** The transformation of the HD signal into a spatial code, although apparently spurious, begs the question of whether the brain may take advantage of this information to generate actual spatial information. In three animals, HD cells were recorded simultaneously in the ADn and the PoSub (Fig. 1a). Two HD neurons from the ADn and the PoSub, with virtually the same preferred direction (toward "South"), showed a similar non-homogeneous firing in the arena, concentrated along the walls parallel to their preferred direction (Fig. 1b, b′). However, a careful examination of the PoSub neuron revealed that the cell fired mostly along the East wall, and not the West wall, unlike what would be expected from a "pure" HD neuron. As a result, this neuron conveyed twice as much uncorrected spatial information than the simultaneously recorded HD neuron from the ADn (Fig. 1c, c′). The difference in uncorrected spatial information conveyed by the two neurons corresponds to the amount of unbiased information of the PoSub

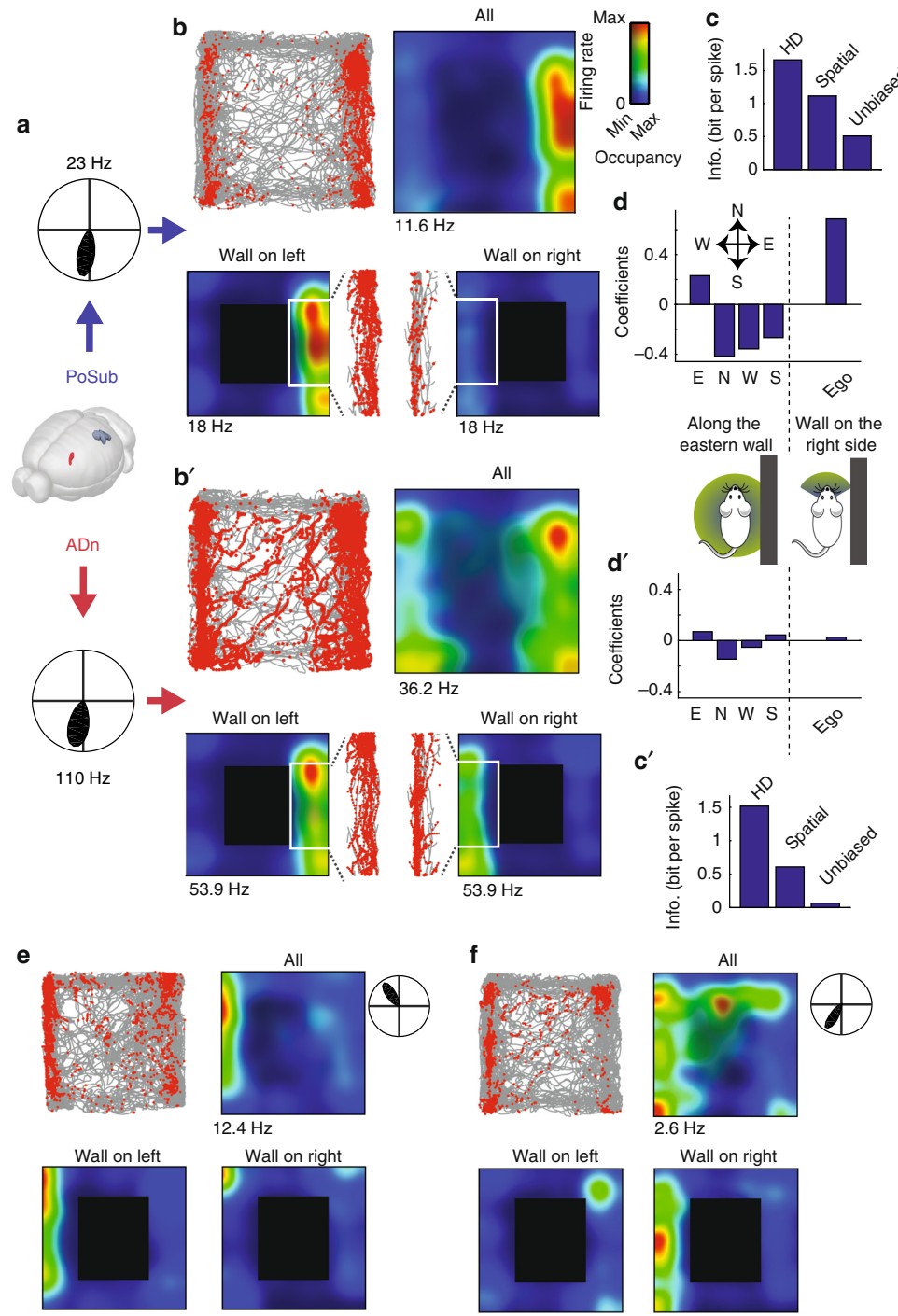

**Fig. 1** Spatial correlates of example HD neurons. **a** Two example HD neurons recorded simultaneously in the PoSub (top) and in the ADn (bottom). Polar plots indicate average firing rate in function of animal's orientation. **b** Top left: position of the animal (gray) superimposed with location of the animal when the neuron spiked (red dots); top right, spatial tuning of the neuron where average firing rate at each location is represented as a colormap. Contrast displays the overall occupancy at each location. Bottom, place fields of the neuron during exploration along the borders when the walls are located on the left of the animal (left panel) or on the right (right panel). Inset show animal's position and spike location in the two configuration during which a neuron governed only by its HD tuning curve is expected to fire. **c** HD, spatial and unbiased information per spike of the neuron shown in **a** (see Methods and Supplementary Fig. 1). **d** Top, regression coefficients of the generalized linear model applied to the binned spike train of the neuron displayed in **a**, consisting of walls (East, North, West or South) or relative wall position (Ego), both regression included expected instantaneous firing rate from HD tuning curve (see Supplementary Fig. 4). Bottom, schemas depicting the two types of behavioral variables used to regress the neuronal data on: left, specific border sensitivity (East, North, West or South) independently of animal's head-direction; right, animal's position relative to the closest wall ('all on the right' or 'wall on the left') within a ± 60° range of head direction. **b'**–**d'** Same as **b**–**d** for a ADn HD neuron. **e**–**f** Additional examples of two wall-modulated HD neurons in the PoSub. Brain displayed in **a** © 2015 Allen Institute for Brain Science. Brain Explorer. http://mouse.brain-map.org/static/brainexplorer

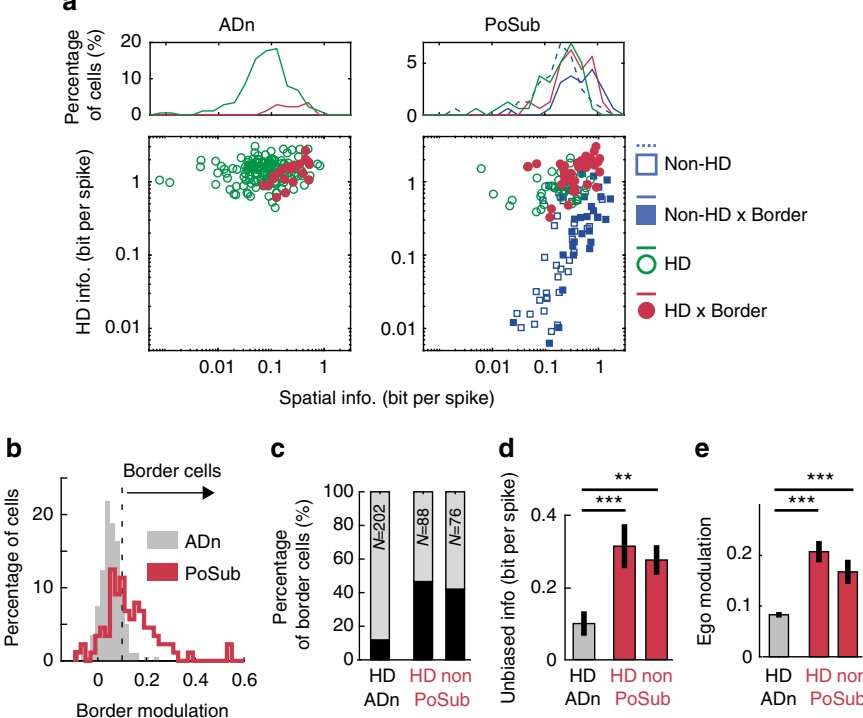

**Fig. 2** Distribution of behavioral correlates in ADn and PoSub cell populations. **a** Bottom, the amount of head-direction vs. unbiased spatial information is shown for all the HD neurons in the ADn (left) and HD and non-HD putative pyramidal neurons in the PoSub (right). HD cells are shown in green, border-modulated neurons as filled markers and HD-by-border as red filled markers. Top histograms display the marginal distribution of unbiased spatial information within each category: green for non-border HD neurons, red for HD-by-border neurons, dashed blue line for non-border, non-HD pyramidal neurons, plain blue for pyramidal border, non-HD neurons. **b** Distribution of border modulation coefficients for HD cells in the ADn (gray bars) and the PoSub (red line). The vertical dashed line indicates the threshold that was chosen to distinguish between border and non-border modulated neurons. **c** Proportion of border-modulated neurons for all cell categories (black bars). Numbers indicate the total number of neurons in each category ('non' stands for 'non-HD'). **d** Average (±s.e.m.) unbiased information in ADn HD neurons and PoSub HD and non-HD pyramidal neurons. **e** Same as **d** for egocentric modulation of neurons. **p < 0.01, ***p < 0.001

HD neuron. Place-by-HD neurons were already reported in the PoSub[33]. However, in our case, the neuron did not show an actual 'place field'. Instead, its firing was suppressed along the West wall where it should have fired had it been a "pure" HD neuron.

We hypothesized that this neuron integrated the HD signal with specific sensory information, hence was biased by behavioral and environmental features. We tested this hypothesis with a Generalized Linear Model[52,53] by regressing the spike trains of the HD neurons separately against each one of the 4 walls, in addition to the expected firing rate from the HD tuning curves (see Methods; Supplementary Fig. 4). Unlike the example ADn HD neuron, the PoSub HD neuron was positively modulated by the presence of the animal along the East wall and 'suppressed' along all other walls (Fig. 1d, d'). This extra sensory modulation may reflect input from a spatially localized cue or could result from a combination of the HD signal with an 'egocentric'-related information, that is the position of environmental elements in the animal's body-centered reference frame[54]. Separating the firing maps of the two example HD neurons when the animal ran left, or right, to the walls showed that the PoSub HD neuron fired preferentially for the former condition, while the ADn neuron fired similarly in both cases. Thus, we also regressed the example neuron's firing pattern with a signal informing whether a wall was situated on the right or left side of its body. This analysis revealed that the firing of the PoSub neuron was better predicted by the combination of HD and a wall being on the left side of the body (Fig. 1b'), and this HD-by-border modulation was observed in other PoSub neurons (Fig. 1e, f)

**Modulation by borders in the post-subiculum**. We then examined how HD and spatial (in particular border) information were distributed in the ADn and PoSub populations. The distribution of unbiased spatial vs. head-direction information did not show particular correlations between the two measures (Fig. 2a), suggesting that external factors, and not higher head-direction information, contribute to the increase of spatial information in certain HD cells. The examples presented in Fig. 1 show that HD cells of the post-subiculum can be modulated by specific borders in the environment. Using the same regression analysis as in Fig. 1, cells were additionally classified as border-modulated when the maximum regression coefficient against border identity (corrected for HD tuning biases) was greater than 0.1 (Fig. 2b). PoSub HD cells showed a stronger modulation by borders than ADn HD cells ($p = 1.7 \cdot 10^{-4}$; $n = 204$, 88 ADn HD neurons and PoSub HD neurons, respectively; Mann–Whitney U test). This modulation by border identity was not a by-product of spurious regression to random variables (Supplementary Fig. 4). However, not all HD neurons in the PoSub were modulated by border and egocentric information. Overall, the effect is continuously distributed from 'pure' HD cells (green dots in Fig. 2b, left panel) to border-modulated HD neurons (red dots; see also Supplementary Fig. 5). Finally, putative pyramidal cells in the PoSub were identified on the basis of their waveform features (see Methods). Approximately 40% of HD and non-HD pyramidal neurons were modulated by borders in the PoSub, only 10% of HD cells in the ADn (Fig. 2c). HD and other pyramidal neurons in the PoSub conveyed an equivalent amount of unbiased spatial

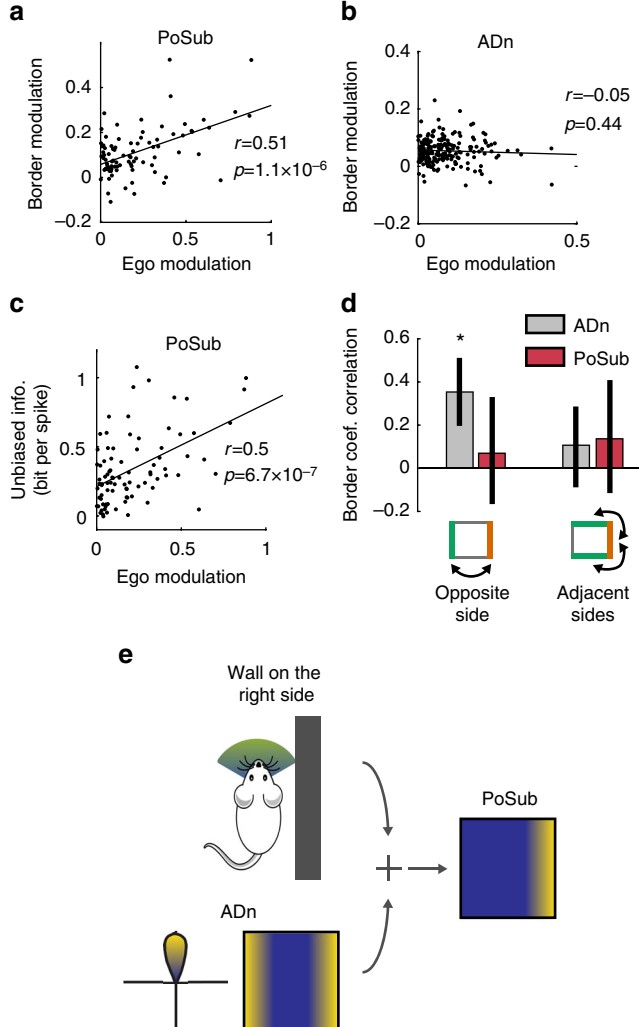

**Fig. 3** HD neurons in the PoSub, but not in the ADn, integrate both allocentric (vestibular) and egocentric (self-centered) information to build an unbiased spatial code. **a** Egocentric modulation of HD neurons in the PoSub correlate with border modulation in for PoSub HD neurons. **b** Same as **a** for HD cells of the ADn. **c** Correlation between egocentric modulation of HD neurons in the PoSub and the amount of unbiased spatial information. **d** Correlation of the maximal border modulation coefficient of each neuron with the coefficient associated with the opposite wall (left) or with the adjacent walls (right; the average of the two border coefficients is considered in this case). **e** Schematic diagram of the transformation of an allocentric HD signal into a spatial code for border by a combination with an egocentric signal

information (Fig. 2d; $p = 0.2$; Mann–Whitney U; $n = 88, 76$ PoSub HD neurons and non-HD pyramidal neurons respectively) but both cell categories conveyed more unbiased spatial information than ADn neurons ($p < 10^{-10}$ for both comparisons, $n = 204$ ADn HD neurons). This higher spatial modulation did not result from difference in firing rates (Supplementary Fig. 6a) or difference in spatial sampling, as in sessions where ADn and PoSub neurons were recorded simultaneously, the average unbiased information of HD neurons was higher in the PoSub in all but one sessions (Supplementary Fig. 6b). PoSub neurons showed also significantly stronger modulation by their relative position to the walls than thalamic ADn neurons (Fig. 2e; ADn HD neurons vs. PoSub HD neurons $p < 10^{-6}$ and non- HD neurons $p = 0.0014$; $n = 204, 88,$ 76 ADn HD neurons, PoSub HD neurons and non-HD neurons respectively; Mann−Whitney U test).

We further tested how these neuronal correlates of spatial and egocentric parameters covaried. Overall, the modulation of HD cells by borders in the PoSub was correlated with their wall-body modulation (Fig. 3a; $r = 0.51$, $p = 1.1 \times 10^{-6}$, Pearson's test), as expected from a combination of HD and egocentric information. This was not the case in the ADn (Fig. 3b; $p > 0.05$). As a result, unbiased information was also correlated with the modulation by wall-body (Fig. 3c; $r = 0.5$, $p = 6.7 \times 10^{-7}$). Finally, we asked how border coefficients correlate with each other. In line with previous results, this analysis shows that for HD cells of the ADn the maximum border coefficients are positively correlated with the coefficients associated with the opposite wall ($p = 2.8 \times 10^{-7}$, Pearson's test), not for adjacent walls (Fig. 3d). This was not the case for HD cells of the PoSub, in either conditions. These results suggest that the combination of a 'pure' HD signal, likely arising from the ADn, and an additional source of information corresponding to the egocentric position of the animal relative to the walls in the environment forms the building block of a spatial signal (Fig. 3e), that could potentially be transformed into a code for the borders of the environment[37,38,40].

**Pure HD cells are rigidly coordinated.** HD neurons in the PoSub that are modulated by spatial factors are modulated not only by the one-dimensional HD signal but are expected to be influenced by other high-dimensional inputs as well. They are thus less likely to exhibit invariant coordination across brain states, especially during sleep when the system is disengaged from its external inputs[21]. To test this hypothesis, we compared pairwise correlations of HD cells in wake and sleep (either Rapid Eye Movement − REM − or non-REM sleep stages). PoSub HD neurons conveying low unbiased spatial info (cell pairs of neurons within the first 33th percentile) had highly preserved correlations during wake and sleep (Fig. 4a, b). In contrast, HD cell pairs that were the most spatially tuned (top 33th percentile) had much less wake-preserved correlations in both REM and non-REM (Fig. 4a, b; Fischer's test, $p < 0.01$ for wake vs. REM and non-REM, $n = 35$ and 37 cell pairs in the bottom and top 33th percentile, respectively). ADn HD neurons showed an almost perfectly preserved correlations across brain states (Fig. 4b; $p > 0.05$; $n = 164$ and 202 pairs of bottom and top 33th percentile, respectively) in agreement with the hypothesis that neurons of the HD circuits are coordinated by internal dynamics that are largely independent of brain state[21].

## Discussion

We demonstrate here that HD cells can carry spurious spatial information under environmental constraints. HD neurons in ADn fired more strongly along the two walls parallel to their preferred direction. The symmetry was broken in PoSub neurons by the addition of sensory information so that the place fields met the criteria of "border cells" firing most effectively close to one of the walls. Our findings demonstrate how the combination of the HD signal and egocentric information leads to a spatial signal.

If the animal's head orientation were unrestricted and homogenous at all locations, the HD system would convey a pure head direction signal, irrespective of the animal's position. However, most animals live in structured environments and possess genetically determined stereotypical behaviors; for example, rodents are agoraphobic and preferentially walk along borders and sheltered routes[49,55]. When environmental signals are combined with HD signals, the downstream "reader" networks can establish a probabilistic inference about the animal's position. Since firing of HD neurons is controlled mainly by distal cues[56], whereas local cues can inform the brain about nearby walls, the

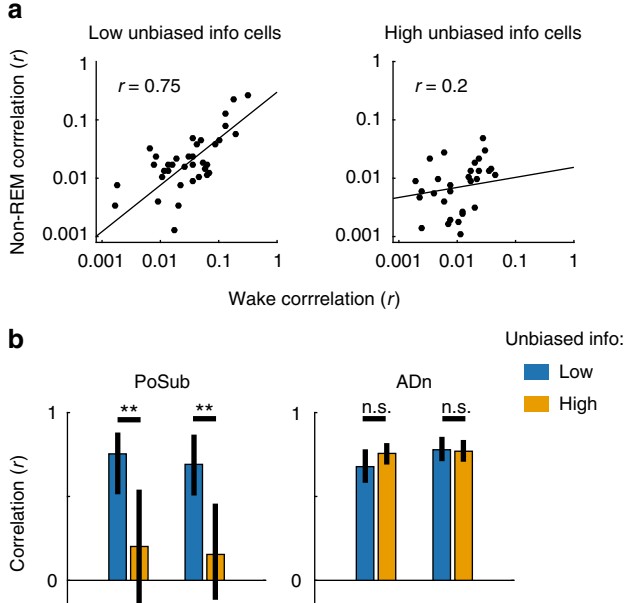

**a**

Low unbiased info cells | High unbiased info cells

$r = 0.75$ | $r = 0.2$

(Non-REM correlation ($r$) vs Wake correlation ($r$))

**b**

Unbiased info: Low (blue), High (orange)

PoSub | ADn

Correlation ($r$)

NREM vs. WAKE | REM vs. WAKE

**Fig. 4** Functional connectivity of spatially related cells is not rigid. **a** Left, pairwise neuronal correlation during non-REM vs. wake for PoSub cell pairs with low unbiased spatial information (bottom 33% percentile); right, same for cell pairs conveying high unbiased spatial information (top 33% percentile). **b** Left, cross-brain states correlations (non-REM vs. wake and REM vs. wake) in the PoSub for neuron pairs conveying low (blue) and high (orange) unbiased spatial information; right, same for ADn cell pairs. **p < 0.01; error bars indicate 95% confidence interval

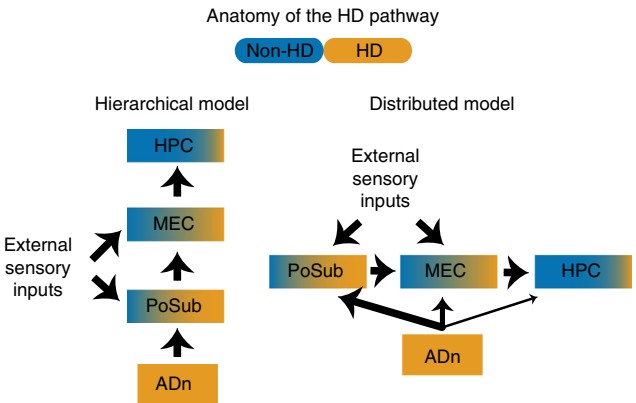

Anatomy of the HD pathway

Non-HD | HD

Hierarchical model | Distributed model

HPC

MEC

External sensory inputs

PoSub

ADn

External sensory inputs

PoSub | MEC → HPC

ADn

**Fig. 5** Anatomical organization of neural information relevant to navigation. Schematic diagram of HD signal transformation to spatial information in a hierarchical (left) or distributed (right) fashion

combined information can be used to represent environmental boundaries.

The definition of borders is crucial in calibrating a spatial representation[9,47,48,57–61]. In addition, borders serve as a "reset" signal to correct for accumulated errors of path integration[62,63]. "Border cells" or "boundary vector cells" in the medial entorhinal cortex and surrounding areas are active along a specific wall of the enclosure[37,38,40]. The origin of this code remains unknown but it is possible that border cells primarily rely on low-level sensory signals: border cells and HD cells emerge early in development, prior to grid cells and place cells[42–46].

Local sensory inputs can be provided by activation of the whiskers (haptic flow) or optic flow near walls. We observed that the association of the HD signal with local egocentric information[54] can break symmetries imposed by the environment[47,48]. Active sensing with whiskers can generate a distance code of the nearest wall in the barrel cortex[64]. As a result of symmetry breaking, spatial information can be incorporated by HD neurons in the PoSub[33]. A "pure" HD cell in a square environment tends to fire along the two opposite walls parallel to its preferred direction. However, when its HD tuning is combined with an egocentric signal, the cell fires only along one of the walls, effectively doubling its spatial information content (as in the example neuron in Fig. 1a–c compared to a′–c′). The combination of these two streams of inputs, HD and local egocentric, may be considered a building block of a code for environment boundaries. When the walls are not parallel to the preferred direction of HD neurons, for example in circular environments, such a mechanism may fail[38]. HD neurons of the PoSub can also be influenced by hippocampal outputs or other sensory inputs and may thus combine HD and place signals. In a circular arena or any arbitrarily shaped environment, the combination of these

signals would still play a key role in disambiguating position based on the HD signal. Combining the HD code with a signal about the egocentric position of the border can disambiguate between parallel portions of the boundaries, irrespective of the shape of the enclosure. It is possible though that the nature of this code depends on the configuration of the environment in which the animal is raised.

ADn neurons, and member cells of the HD circuit in general, are believed to be endowed with attractor dynamics, emerging from local circuits or inherited from upstream attractors[11,13,65]. This hypothesis was confirmed by the demonstration that temporal coordination between HD neurons was strongly preserved across brain states, suggesting a system largely driven by internally generated activity[21]. In contrast, spatially tuned HD neurons that are driven by external signals may reduce their temporal coordination in other behavioral contexts, for example during sleep. In agreement with this hypothesis, the coordination between HD neurons in the PoSub that were the most spatially modulated was much less preserved during sleep compared to wake than in HD neurons with no or limited spatial modulation. These findings point to the relatively 'rigid' and 'plastic' properties of the members of the HD and spatial systems, respectively.

The organization of the HD circuit is often described as a hierarchical, bottom-up network where the HD signal is transmitted from the ADn to the hippocampus via multiple synapses[11,13] (Fig. 5). However, the ADn projects to multiple locations of the parahippocampal formation, including the parasubiculum, dorsal subiculum and the medial entorhinal cortex[20,22]. It is possible that the hippocampus is also directly reached as HD neurons have been reported in the CA1 region[16,27,28]. The HD circuit would be thus best described by a distributed pathway, with ADn playing the role of a central hub for HD information (Fig. 5). In addition, our findings indicate that certain HD cells in the PoSub may convey more information beyond the HD signal[33]. Similarly, some neurons of the medial entorhinal cortex exhibit a conjunction of HD and grid signals[29,32]. More generally, neurons in the medial entorhinal cortex and the subicular complex show a wide range of combination of spatial, HD and speed signals[29–36,39]. The HD signal present in many neurons of these structures can ultimately support a spatial map or a path integration process.

HD information is essential for maintaining grid cells of the medial entorhinal cortex[15] and may also be important for hippocampal place cells. Although HD and spatial signals get combined in several structures, the two streams of information appear to keep their identity. Neurons with true spatial properties are modulated by hippocampal theta oscillations (Brandon et al.[32])

and neurons modulated by the theta rhythm provide a dynamical definition of the limbic system[66]. In contrast, 'pure' HD neurons, independent of the structure in which they reside, are not affected by theta oscillations[32,33]. Instead, members of the HD system communicate by a fast oscillation (150–250 Hz)[21]. These separate streams may serve to distinguish between externally imposed and internally coordinated mechanisms.

## Methods

**Electrodes and surgery and data acquisition.** All experiments were approved by the Institutional Animal Care and Use Committee of New York University Medical Center. Some of the data from the present report were previously published (Peyrache et al.[21]). Briefly, seven mice were implanted with silicon probes (4, 6, or 8 shank, 32 or 64 channel Buzsaki probes, Neuronexus, MI) in the ADn (coordinates from bregma: Antero-Posterior: −0.6 mm; Medio-Lateral:−0.5 to −1.9 mm; Dorso-Ventral: 2.2 mm). Three out of these seven mice were also implanted over the post-subiculum (coordinates from bregma: AP: −4.25 mm: ML: −1 to −2 mm; DV: 0.75 mm).

During the recording session, neurophysiological signals were acquired continuously at 20 kHz on a 256-channel Amplipex system (Szeged; 16-bit resolution, analog multiplexing). The wide-band signal was downsampled to 1.25 kHz and used as the LFP signal. For tracking the position of the animals on the open maze and in its home cage during rest epochs, two small light-emitting diodes (5-cm separation), mounted above the headstage, were recorded by a digital video camera at 30 frames per second. The LED locations were detected online and resampled at 39 Hz by the acquisition system. Spike sorting was performed semi-automatically, using KlustaKwik (http://klustakwik.sourceforge.net/). This was followed by manual adjustment of the waveform clusters using the software Klusters.

After 4–7 d of recovery, probes were lowered towards their target structures. In animals implanted over the ADn, the thalamic probe was lowered until the first thalamic units could be detected on at least 2–3 shanks. At the same time, hippocampal wires were slowly lowered until reaching the CA1 pyramidal layer, characterized by high amplitude ripple oscillations. The thalamic probe was then lowered by 70–140 μm at the end of each session. In the animals implanted in both the thalamus and in the PoSub, the subicular probe was moved everyday once large HD cell ensembles were recorded from the thalamus. Thereafter, the thalamic probes were left at the same position as long as the quality of the recordings remained. They were subsequently adjusted to optimize the yield of HD cells. To prevent statistical bias of neuron sampling, we discarded from analysis sessions separated by less than 3 days during which the thalamic probe was not moved.

**Statistics.** Sample sizes (number of animals and number of neurons) are similar to those reported in previous publications, but no methods were used to predetermine sample sizes. We used non-parametric statistical methods everywhere unless stated otherwise.

**Recording sessions and behavioral procedure.** Recording sessions were composed of exploration of an open environment (wake phase) followed and preceded by rest/sleep epochs. Animals implanted in the ADn were foraging for food for 30–45 min in a 53- × 46-cm rectangular arena surrounded by 21-cm-high, black-painted walls on which were displayed two salient visual cues. A total of 37 sessions showing homogeneous visit within the environment were analyzed. In 19 sessions (three animals), the ADn and PoSub were simultaneously recorded. Two of the four only animals implanted in the ADn (not the PoSub) explored a radial arm maze (three sessions each) in addition to the open environment (Supplementary Fig. 3). All experiments were carried out during daylight in normal light-dark cycle.

All behavioral measures were computed when the animal was in movement (speed >2.5 cm/s)

**Head-direction neurons.** HD tuning curves were obtained by counting the number of spikes for each head direction (in bins of 6°) and then divided by the occupancy (in seconds) of the animal in each direction bin. Neurons were categorized as HD cell when a shuffling of their spike train resulted in a significant resulting vector (1000 shuffles, $p <= 0.001$), when they showed high stability between the first and second half of the session (Pearson's correlation between the tuning curves r > 0.75) and the concentration factor of the spikes' phases around their mean of the tuning curve was higher than 1 (Fisher, 1993).

**Information measure.** Spatial (or HD) information was first computed as the mutual information between animal's location (or HD) and the neurons' firing rate at each location[67].

$$ I = \sum_x \lambda(x) \log_2 \frac{\lambda(x)}{\lambda} p(x) $$

where $x$ is either a spatial or head-direction bin, $\lambda(x)$ is the firing rate of the

neuron in the bin $x$, $\lambda$ is the mean firing rate and $p(x)$ is the probability of occupancy at bin $x$.

Spatial mutual information decreases monotonically with the size of spatial bins. To control for this confound, as shown in Supplementary Fig. 2, spatial information was estimated through a cross-validation procedure[52]. In total 90% of the data were used to estimate the "place field" of the neuron (training data). This tuning curve was then used to predict the instantaneous firing rate of the neuron during the remaining 10% of the data (test data). Spatial binning was fixed (0.5 cm) and place field computed from the training data were smoothed with Gaussian kernels of different spatial variance. Spike trains that showed at least a maximum within the range of spatial smoothing (0.5–25 cm) were best predicted with a spatial scale of 2–4 cm.

**Unbiased spatial information.** A total of 500 control spike trains were generated for each neuron as random Poisson process depending only on the HD tuning curve of the actual neuron and instantaneous heading of the animal (Supplementary Fig. 1). The difference between the observed spatial information and the average spatial information from the random spike trains was defined as the unbiased information. This indicates the amount of spatial information that did not depend on the HD tuning of the neuron. This method is essentially the same as the original method proposed by Muller et al.[51]. The method developed by Acharya et al.[16] is based on a direct regression (with a Generalized Linear Model) on a set of functions of angle and space. However, this decomposition works only in circular environments. The information-metric approach of Burgess et al.[50] is theoretically the most accurate but requires a high sampling in place-by-head-direction that is hard to achieve for sessions of limited duration (even for 30 min of exploration in a $50 \times 50$ cm$^2$ box). This is why we have kept the analysis of each contribution "separated".

**Behavioral bias.** For each position, the histogram of orientation was computed as the histogram of the double values of animal's heading. This doubling transforms direction into orientation, e.g., 90° and 270° are, when doubled, both equal to 180° (modulo 360°). Average orientation values were computed as the direction of the resulting vector. Concentration factor was estimated by assuming the distribution followed a Von Mises' distribution.

**Classification of PoSub neurons.** As previously reported[21], putative interneurons and pyramidal cells can be discriminated by clustering their action potential waveforms. Putative pyramidal cells were characterized by broad waveforms, whereas putative interneurons had narrow spikes. To separate between the two classes of cells, we used two waveform features: (i) total duration of the spike, defined as the inverse of the maximum power associated frequency in its power spectrum (obtained from a wavelet transform) and (ii) the trough-to-peak duration. Putative interneurons were defined as cells with narrow waveform (duration <0.9 ms) and short trough to peak (<0.42 ms). Conversely, cells with broad waveforms (duration >0.95 ms) and long trough-to-peak (>0.42 ms) were classified as putative pyramidal cells.

**Generalized linear model.** Binned spike trains were regressed with different observables using a logarithmic 'link' function, assuming that spike trains are Poisson processes[52,53]. First, spike trains were binned in windows of 5 ms. The resulting vectors were smoothed with a Gaussian filter (s.d. 25.6 ms, corresponding to the video sample duration). Then, to estimate the modulation by borders, binned spike trains were regressed on four binary variables indicating the presence of the animals along each of the wall (at less than 15 cm). The four corners were excluded. 'Border modulation' was defined as the maximal regression coefficient among the four coefficients associated with each wall. To estimate how this modulation was independent from spiking directly explained by the HD tuning curve, the neurons were additionally regressed on the instantaneous firing rate expected from the current heading of the animal and the neuron's HD tuning curve (see Supplementary Fig. 4).

To estimate the modulation by egocentric signals, binned spike trains were regressed on position of the animal relatively to the walls, defined as two binary values indicating whether the nearest wall (at most 15 cm away) was on the right or left side of the animal (±60°). "Egocentric modulation" was defined as the absolute difference between the regression coefficients associated with the wall on the right or the left of the animals. Like the modulation by border, binned spike trains were additionally regressed on the instantaneous firing rate drawn from their HD tuning curve.

To normalize the contribution of each variable, they were all z-scored (binary variables describing proximity of a wall or body position relative to a wall, as well as expected firing rate). In addition, binned spike trains were regressed on an additional constant term (of constant value). This is equivalent to making the resulting coefficients independent of neurons' firing rates.

Mathematically, this corresponded to determine the vector of unknown parameters $\beta$ that best explained the firing rate $\lambda_i(t)$ of cell $i$ at time $t$ through the

relation:

$$\lambda_i(t) = \exp\left(\beta_{i,0} + \sum_{k=1}^{N} \beta_{i,k} X_k(t)\right)$$

where $X_k(t)$ is the (z-scored) value of the $k$th observable (out of N) at time $t$, the term $\beta_{i,0}$ captures the average activation of the neuron (equivalent to a parameter associated with an observable of constant value) and $\beta_{i,k}$ is the parameter associated with the $k$th observable for neuron $i$.

**Code availability**. Computer code is available online (TSToolbox: https://github.com/peyrachelab) and on demand.

**Data availability**. Data are available at http://crcns.org/data-sets/thalamus/th-1 (doi:10.6080/K0G15XS1).

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

## Acknowledgements

We thank Lisa Roux for comments on this manuscript. The work was supported by US National Institute of Health grants NS34994, MH54671, and NS074015. A.P. was supported by a National Institute of Health Award K99 NS086915.

## Author contributions

A.P. and G.B. designed the experiments. A.P. conducted the experiments. A.P. designed and performed the analyses with the help of N.S. A.P. and G.B. wrote the paper.

## Additional information

**Competing interests:** The authors declare no competing financial interests.

