## [Peer Review File · Nature Communications]

Reviewers' comments:

Reviewer #1 (Remarks to the Author):

The stated goal of this study is to understand how the head direction signal is transformed in to allocentric spatial code. To achieve this goal, the authors measured the modulation of neural activity in several different brain areas. The key novel claim is that head direction selectivity, combined with behavioral bias could generate allocentric spatial selectivity and this is found to occur more in post subiculum than in ADn.

This fundamental premise of the study is difficult to reconcile. Neurons in hippocampal system show allocentric spatial selectivity even when the mouse's spatial behavior is random. But, the proposed mechanism of allocentric selectivity relies heavily on behavioral bias and may even be absent without behavioral biases. This mechanism cannot explain the commonly observed hippocampal allocentric spatial selectivity.

In general, it seems difficult to define behavioral bias based spatial maps. Such 'allocentric' maps will be found in any part of the sensory or motor cortex, which seems like an artifact. The behavioral bias based maps will be quite unstable, and change substantially as the subject's behavior changes, making these maps of unreliable for allocentric information.

The data are interesting but this fundamental assumption seems difficult to accept. The authors should factor out the behavioral bias before estimating neural selectivity, as done commonly. The key results about HD selectivity are packed in just one figure and difficult to parse. What fraction of neurons showed significant spatial and head directional modulation in different brain areas? This information needs to be clearly shown in figures.

"...Various solutions have been suggested to separate the independent contribution of place and HD tuning to the firing of a HD neuron (Burgess et al., 2005; Muller et al., 1994). However, those methods usually require high spatial sampling...." But, the authors cite Acharya et al. (Cell 2016), which did not mention such difficulties. They seem to have estimated independent contribution of spatial and directional signals to neural responses, without any behavioral bias. Why can't the same methods be applied here? What are the similarities and differences, dis/advantages of the proposed method compared to the earlier method? The proposed method requires Poisson spiking activity, generated based on certain head direction tuning curve. This sounds circular since estimation of the head direction tuning curve of the neuron is unknown and the purpose of the method is to obtain such an estimate?

"...HD neurons in ADn showed symmetric, elongated 'place fields'..". This is difficult to accept because place field is thought to be largely independent of behavioral biases. While there are notable exceptions to this rule, place field or allocentric selectivity needs to be demonstrated independent of behavioral bias.

Reviewer #2 (Remarks to the Author):

Peyrache et al. compare the firing properties of antero-dorsal thalamic (ADn) neurons and post-subiculum (pSUB) neurons according to their correlates to environmental borders, environmental locations, and head orientations. The main conclusion of the work is that pSUB neurons exhibit strong tendencies to combine head orientation with proximity to specific borders while ADn neurons exhibit purer head-direction firing correlates. The results are interpreted according to the known roles of pSUB and ADn neurons in coordinating head-direction selective firing in other structures and according to the potential role of pSUB in generating spatial firing correlates (place cells) in hippocampus.

The firing dynamics within each region are well-analyzed such that I have few qualms with the validity of the contrasts made between ADn and pSUB neurons. The sensitivity of the analysis to the way the environment is sampled is impressive. The work is likely to have a strong impact on the large field of researchers examining the many forms of space and orientation dependent firing that exist in hippocampal and parahippocampal structures and related cortical regions. Nevertheless, the manuscript, in its present form, falls somewhat short of forming a coherent story that will change the way the field thinks about pSUB in particular. In my opinion, this could be a relatively important contribution if the authors can address the following issues:

1. One minor, but rather intrusive, problem with the manuscript is that most of it is taken up with comparison of firing properties for ADn and pSUB but that it ends with a burst of small pieces of data concerning head direction tuning of a small population of CA1 neurons. This component is distracting and anecdotal and fails to add much to the picture. I think the manuscript would be much better without it at all.
2. Again, a minor point – In the introduction, the authors refer to weaknesses in assessing spatial firing in head direction sensitive neurons that concern the animal's inability to take certain orientations in certain parts of the environment or the animal's bias toward traveling in certain directions in certain parts of the environment. I appreciate the reference to animals' tendencies to "wall-follow", but simply referring to stereotypical behaviors doesn't clearly communicate what's really happening in a simple fashion.
3. Line 148 – The reference to the Wilber paper here is rather obscure. Be more specific as to what is meant by "egocentric" information.
4. Line 296 – "...maintaining grid cells of the entorhinal grid cells..." Please rephrase.
5. Line 267 – Can the authors flesh out what is meant by saying that the building blocks for a code of environment boundaries may fail in a circular arena. Is it really the case that border-specific firing is much worse in circular arenas? Also, the authors may take a moment to think about how this may play out in more complex environments. The reality is that rats are rarely charged with encoding position relative to an arena of any shape (except in laboratories).
6. The authors make a strong case for pSUB neurons having strong spatial components in their firing patterns. In a sense, the case is so strong that the reader, especially given the examples presented, may be left to wonder whether head direction contributes very much to pSUB firing. In figure 1, for example, the pSUB neuron's firing is almost non-existent when the animal moves through the arena's central regions and is VERY weak along the west border once there is an accounting of occupancy time. In such a case, what is the strong case for referring to the neuron as a head direction cell? The authors might do well to provide more examples of pSUB neurons that give a better flavor for how distributed the responses are between purely spatial, purely head direction, and purely egocentric

(border on left versus right). This, in my opinion, would go a long way toward allowing the field to think more broadly about the function of this important region (which is mostly known for head direction cells).

7. The authors address the spatial component through the distinction between border walls (and their coefficients in the models). But to what extent does the data reflect a border, categorically (ESN or W), versus an allocentric space? Have the authors considered the relationships between coefficients? For example, strong positive coefficients for N and W might indicate a position in space or presence of a corner as opposed to a single border while a preponderance of neurons having positive coefficients for one border, but negative coefficients for the remaining borders would indicate encoding of specific borders as opposed to place per se. The point here is to say as much as possible about the nature of the "spatial" sensitivities of pSUB neurons. In this respect, the authors should make the best attempt possible to reference interpretations of pSUB firing patterns in already-published data.

We would like to thank the reviewers for their very valuable comments and suggestions. Our point-by-point responses are below. Following the reviewers' suggestions, we have made several modifications to our manuscript (highlighted in red). Most importantly, we have included a new analysis of the population correlates to space, and split the former Figure 2 in two new figures. We hope the reviewers will agree that with our responses and modifications the manuscript improved significantly.

Reviewer #1 (Remarks to the Author):

The stated goal of this study is to understand how the head direction signal is transformed into allocentric spatial code. To achieve this goal, the authors measured the modulation of neural activity in several different brain areas. The key novel claim is that head direction selectivity, combined with behavioral bias could generate allocentric spatial selectivity and this is found to occur more in post subiculum than in ADn.

This fundamental premise of the study is difficult to reconcile. Neurons in hippocampal system show allocentric spatial selectivity even when the mouse's spatial behavior is random. But, the proposed mechanism of allocentric selectivity relies heavily on behavioral bias and may even be absent without behavioral biases. This mechanism cannot explain the commonly observed hippocampal allocentric spatial selectivity.

In general, it seems difficult to define behavioral bias based spatial maps. Such 'allocentric' maps will be found in any part of the sensory or motor cortex, which seems like an artifact. The behavioral bias based maps will be quite unstable, and change substantially as the subject's behavior changes, making these maps of unreliable for allocentric information.

The data are interesting but this fundamental assumption seems difficult to accept. The authors should factor out the behavioral bias before estimating neural selectivity, as done commonly.

We agree with the referee that many 'neuronal codes' (or more precisely correlations) across sensory systems likely convey a high amount of spurious spatial information. However, a key issue is how does a 'cognitive map' arise from such sensory inputs (or at least be affected by them) and how sensory inputs affect remapping in different environments. A computational modelling assumption is that boundary vector cells (or border cells) are critical for the formation of hippocampal place cells. In this work, we raise the possibility that boundary vector cells or border cells are in fact a combination of sensory inputs and environment confined behavior. There is no contradiction between a stable and abstract hippocampal map and a

behavior-dependent modulation of upstream neurons. What we demonstrate is that neurons at increasing 'distance' from the sensory inputs have increasing share of internally-constructed patterns. According to our null hypothesis, sensory inputs should exert similar effects on simultaneously recorded HD neurons in the ADn and the PoSub. But they do not. They convey different amount of 'biased' or 'actual' spatial information.

Our findings clearly show that PoSub cells also incorporate non-HD information that do increase the amount of spatial information, which cannot simply be explained by HD tuning and behavior. PoSub HD neurons conveyed more 'unbiased' spatial information than their ADn counterpart.

Importantly, our findings demonstrate that when behavior is consistent or consistently biased, as it occurs in environments with walls, the combination of behavior and HD information can generate border cells. What appeared interesting to us is that this HD-derived border 'code' may be a precursor for the emergence of a genuine allocentric representation the environment.

The key results about HD selectivity are packed in just one figure and difficult to parse.

We agree that Figure 1 is 'packed'. It serves to explain our rationale by two simultaneously recorded example neurons in the PoSub and the ADn. (and two other examples from the PoSub). All group analyses are presented in the following figures. If the reviewer believes that we should further simplify the figure, we can move parts of it to the Supplementary section.

What fraction of neurons showed significant spatial and head directional modulation in different brain areas? This information needs to be clearly shown in figures.

This information is now explicitly shown in Figure 2C: Approximately 40% of the HD cells in the PoSub but only 10% are in the ADn are modulated by the walls. We have also added extensive data regarding cell classification in Figure 2A and B. As reported before (Peyrache et al. 2015), and in agreement with the literature, about 30% of PoSub cells were considered HD cells following our 'conservative' criterion. However, as can be seen in Figure 2A, putative pyramidal cells of the PoSub all convey some significant amount of HD info and spatial info.

"...Various solutions have been suggested to separate the independent contribution of place and HD tuning to the firing of a HD neuron (Burgess et al., 2005; Muller et al., 1994). However, those methods usually require high spatial sampling..." But, the

authors cite Acharya et al. (Cell 2016), which did not mention such difficulties. They seem to have estimated independent contribution of spatial and directional signals to neural responses, without any behavioral bias. Why can't the same methods be applied here? What are the similarities and differences, dis/advantages of the proposed method compared to the earlier method?

The proposed method requires Poisson spiking activity, generated based on certain head direction tuning curve. This sounds circular since estimation of the head direction tuning curve of the neuron is unknown and the purpose of the method is to obtain such an estimate?

The referee raises an important point here. Our idea was not to develop yet another method, but we sought a versatile and easy-to-understand approach. In fact, our method is essentially the same as the original method proposed by Muller et al. (1994): in each location bin, the expected firing rate is computed as the number of spikes expected from the HD tuning curve (i.e. for each direction, the time spent times the firing rate from the HD tuning curve for this direction). There are two differences in our approach: instead of using directly the rate of the HD tuning curve to compute the expected rate in space, we generated spike trains and computed 'place fields'. This was made mainly for illustrative purpose (one can see in Supplementary figure 1 that the distribution of spikes in space is exactly the same as the actual neuron) but the results are the same. The second difference is that we compared the resulting place fields with quantitative measures ('information rate').

The method developed by Acharya et al. (2016) is slightly different. They fit the spike train with a Generalized Linear Model: because the number of spikes cannot be linearly related to a raw angle value or a (x,y) spatial position, they transform these values into linear elements: cosine and sine of the angle, and Zernike polynomials to describe the spatial location. It's an interesting approach but their decomposition works only in circular environments. In principle, it can be generalized to arbitrary environment shapes, but that approach would need further justification. In essence, our approach is similar when we use a GLM to linearly regress the neural responses against the expected firing rate from the HD curve and the border locations. The last method that is worth pointing out here is by Burgess et al. (2005): It is a generalization of the information approach we use in this paper. Basically, they tested whether the firing of one neuron is best explained by only its HD tuning curve, its place field or a conjunction of this two behavioral parameters. Theoretically, this method is the most accurate. Unfortunately, it is hard to implement in practice because it requires the evaluation of a 3D place field (place-by-head direction) that is strongly under-sampled for sessions of limited duration (even for 30 minutes of exploration in a 50x50cm box). This is why we have kept the analysis of each contribution 'separated'.

Note that all the above methods assume that neurons are Poisson processes. However, it should be noted that this assumption is not very sensitive to the actual distribution of spikes. Indeed, all these analyses rely only on the average firing rate.

We have now discussed this point in the Supplemental Methods section.

"...HD neurons in ADn showed symmetric, elongated 'place fields'..". This is difficult to accept because place field is thought to be largely independent of behavioral biases. While there are notable exceptions to this rule, place field or allocentric selectivity needs to be demonstrated independent of behavioral bias.

Indeed, this is why we placed the phrase in parentheses. In the revision version, we have changed this sentence to:

"...HD neurons in ADn showed fire more strongly along the two walls parallel to their preferred direction"

Reviewer #2 (Remarks to the Author):

Peyrache et al. compare the firing properties of antero-dorsal thalamic (ADn) neurons and post-subiculum (pSUB) neurons according to their correlates to environmental borders, environmental locations, and head orientations. The main conclusion of the work is that pSUB neurons exhibit strong tendencies to combine head orientation with proximity to specific borders while ADn neurons exhibit purer head-direction firing correlates. The results are interpreted according to the known roles of pSUB and ADn neurons in coordinating head-direction selective firing in other structures and according to the potential role of pSUB in generating spatial firing correlates (place cells) in hippocampus.

The firing dynamics within each region are well-analyzed such that I have few qualms with the validity of the contrasts made between ADn and pSUB neurons. The sensitivity of the analysis to the way the environment is sampled is impressive. The work is likely to have a strong impact on the large field of researchers examining the many forms of space and orientation dependent firing that exist in hippocampal and parahippocampal structures and related cortical regions. Nevertheless, the manuscript, in its present form, falls somewhat short of forming a coherent story that will change the way the field thinks about pSUB in particular. In my opinion, this could be a relatively important contribution if the authors can address the following issues:

We thank the referee for this very kind comment. We have tried our best to improve the quality of our manuscript and we hope that our manuscript improved significantly.

1. One minor, but rather intrusive, problem with the manuscript is that most of it is taken up with comparison of firing properties for ADn and pSUB but that it ends with a burst of small pieces of data concerning head direction tuning of a small population of CA1 neurons. This component is distracting and anecdotal and fails to add much to the picture. I think the manuscript would be much better without it at all.

We respectfully request to retain the hippocampal neurons for the following reasons. The head-direction system has long been described as a 'labeled line' (to paraphrase what usually describes other sensory pathways), that is a hierarchical, feed-forward circuit that gradually transforms a vestibular signal into a place/grid code. We think this is not quite accurate. It is well accepted that the principal source of the HD signal in the cortex originates from the ADn. However, this can be accomplished by two models. One is a serial version with the hippocampus being the last station. The other model is that HD information is broadcasted to multiple downstream targets in parallel. Our findings in the hippocampus strongly support the second model by showing the presence of ADn axons in the alveus close to the subicular border. This is the part of the CA1 subregion which receives mainly entorhinal inputs and generates the most robust place fields. While we agree that more extensive experiments will be needed to map the HD contribution in hippocampal neurons, the present findings are sufficient to support one alternative more than another.

2. Again, a minor point – In the introduction, the authors refer to weaknesses in assessing spatial firing in head direction sensitive neurons that concern the animal's inability to take certain orientations in certain parts of the environment or the animal's bias toward traveling in certain directions in certain parts of the environment. I appreciate the reference to animals' tendencies to "wall-follow", but simply referring to stereotypical behaviors doesn't clearly communicate what's really happening in a simple fashion.

We are glad the referee agree with our reference to animal's behavioral bias. We have clarified this point in the introduction:

"Specifically, we present evidence that, because animal's heading is not homogeneously distributed in space, HD cells in the thalamus (ADn) convey spatial information from the viewpoint of a downstream reader."

3. Line 148 – The reference to the Wilber paper here is rather obscure. Be more specific as to what is meant by "egocentric" information.

We apologize if the context of this reference was not clear. A large body of literature

has investigated 'egocentric' signals in the brain, that is spatial information in the reference frame centered on the animal's position, for example in 'go left or right' exploration task. Various hypotheses regarding the brain systems supporting this type of signal have been put forward but we did not find it necessary to review them extensively as our working hypothesis is rather simple: when the animal is running along a wall, there may be an additional signal from other external sensory inputs that informs the brain that a 'wall is on the right/left of the animal', independent of the absolute position of the animal in its environment. This position-independence is referred to as egocentric. In the Wilber et al. study, the authors described a set of cells in the posterior parietal cortex that code for the 'egocentric cue direction', that is the bearing of a visual landmark relative to the animal's current heading. We used this reference as an example of an 'egocentric' signal in the brain. We have modified this sentence to better explain the meaning of egocentric:

"This extra sensory modulation may reflect input from a spatially localized cue or could result from a combination of the HD signal with an 'egocentric'-related information, that is the position of environmental elements in the animal's body-centered reference frame (Wilber et al., 2014)."

4. Line 296 – "...maintaining grid cells of the entorhinal grid cells..." Please rephrase.

Thank you for spotting this. Corrected.

5. Line 267 – Can the authors flesh out what is meant by saying that the building blocks for a code of environment boundaries may fail in a circular arena. Is it really the case that border-specific firing is much worse in circular arenas? Also, the authors may take a moment to think about how this may play out in more complex environments. The reality is that rats are rarely charged with encoding position relative to an arena of any shape (except in laboratories).

To clarify this, we have added the following sentences at the end of the paragraph:

In a circular arena or any arbitrarily shaped environment, the combination of these signals would still play a key role in disambiguating position based on the HD signal. In fact, the less rectangular is the environment, larger number of HD cells may fire at a given position along the boundaries of the environment. Combining the HD code with a signal about the egocentric position of the border can disambiguate between parallel portion of the boundaries, irrespective of the shape of the enclosure. It is possible though that the nature of this code depends on the configuration of the environment in which the animal is raised.

6. The authors make a strong case for pSUB neurons having strong spatial components in their firing patterns. In a sense, the case is so strong that the reader,

especially given the examples presented, may be left to wonder whether head direction contributes very much to pSUB firing. In figure 1, for example, the pSUB neuron's firing is almost non-existent when the animal moves through the arena's central regions and is VERY weak along the west border once there is an accounting of occupancy time. In such a case, what is the strong case for referring to the neuron as a head direction cell? The authors might do well to provide more examples of pSUB neurons that give a better flavor for how distributed the responses are between purely spatial, purely head direction, and purely egocentric (border on left versus right). This, in my opinion, would go a long way toward allowing the field to think more broadly about the function of this important region (which is mostly known for head direction cells).

We thank the reviewer for alerting us regarding this issue. Indeed, the firing of a few HD cells in the PSub may be strongly modulated by spatial factors, but not all of them. Figures 2A-C now give a more detailed account of the proportion of cells that show such additional correlates. With an arbitrary threshold of border modulation of 0.1 (i.e. the maximal linear regression in a Generalized Linear Model of the neuron's spike train against the presence of one of the walls), approximately 40% of HD and non-HD pyramidal cells in the PSub show a strong modulation by border. In contrast, only 10% show such features in the ADn. However, it would be not correct to claim that these cells do not code for HD (in fact, and HD cells were first described in PSub). The PSub is the main target of the ADn (where, as we show, cells are modulated only by the HD), and a large body of literature has confirmed in many different preparations and species that PSub neurons are bona fide HD cells.

7. The authors address the spatial component through the distinction between border walls (and their coefficients in the models). But to what extent does the data reflect a border, categorically (ESN or W), versus an allocentric space? Have the authors considered the relationships between coefficients? For example, strong positive coefficients for N and W might indicate a position in space or presence of a corner as opposed to a single border while a preponderance of neurons having positive coefficients for one border, but negative coefficients for the remaining borders would indicate encoding of specific borders as opposed to place per se. The point here is to say as much as possible about the **nature** of the "spatial" sensitivities of pSUB neurons. In this respect, the authors should make the best attempt possible to reference interpretations of pSUB firing patterns in already-published data.

We thank the reviewer for this important suggestion. We have now performed this analysis by asking how border coefficients correlate between each other. In line with other analyses, this new analysis shows that for HD cells of the ADn the maximum border coefficients are positively correlated with the coefficients associated with the opposite wall, not for adjacent walls (Figure 3D). A negative correlation between

coefficients associated with adjacent walls may have been expected, but it is not hard to think why it is not the case. This would be the case if cells fire at extremely low firing rates along the adjacent walls, which is only possible if the animal did always run along to these walls. Obviously, this is not true in a real experiment since the mouse sometimes stops and looks around.

We did not find a reliable correlation for HD cells of the PoSub, in either conditions. This shows, once again, that these cells likely receive additional signal coding for the wall identity.

Reviewers' comments:

Reviewer #1 (Remarks to the Author):

The authors have addressed all of my concerns. The revised manuscript is significantly improved.

Reviewer #2 (Remarks to the Author):

The authors have improved the manuscript in many respects. This can be a relatively important paper concerning spatial versus directional tuning in PoSub. As the emphasis in PoSub work has been placed mainly on head direction neurons, this paper could go some way to breaking new ground on the function of this region. However, I still have significant reservations concerning a key aspect of the work (HD tuning in PoSub), how it compares to that observed in prior work (where similar phenomena must be present), and the appropriateness of the data presented in figure 5. I recommend the authors take the opportunity to provide a more definitive account of the firing properties of their PoSub neurons and recommend that figure 5 and associated text be removed.

In the revised manuscript, the authors provide some degree of information concerning the distribution of sensitivity of ADn and PoSub neurons to "spatial" information such as that provided by specific environmental borders. In particular, figure 2B shows that the border modulation scores for PoSub neurons fall on a continuum broader than that for the ADn neurons and with a higher overall mean. But where the PoSub distribution breaks away from the ADn distribution, the border modulation values are already near that for the PoSub neuron depicted in figure 1B. The problem is that I do not see how this neuron, nor the other two example neurons (figure 1E,F) from PoSub, can be clearly characterized as "head direction" neurons in the first place. There is extremely little firing within the central portions of the arena as well as along the west wall. What is there to indicate that the neuron is not responding to the presence of the east wall in close proximity to the left side of the animal? Is the head direction tuning curve not simply an epiphenomenon? I do understand that the authors are trying to say that the PoSub is an area where "spatial" signals, be they allocentric or egocentric, can strongly influence neurons that may be strongly influenced by head direction signals. But the authors should at least acknowledge in discussion that at some point, a neuron should not be recognized as a head direction neuron at all. It might help to include examples of PoSub neurons whose tuning is not strongly affected by border identities or proximities. Do PoSub "head direction" neurons with border modulation scores of less than 0.1 exhibit much firing in the center of the arena or do they simply have more similar firing patterns across parallel borders? If the latter is true, then it would seem that virtually none of the recorded PoSub neurons should be considered head direction neurons, but simply as neurons impacted by several spatial variables with some influence by head direction. The authors stated purpose in publishing this work is to show that PoSub "head direction" neurons are much more "spatial" in their firing patterns and so a more thorough consideration of the true influence of head direction is necessary.

I still find the data of figure 5 to be an unnecessary distraction. As the authors show through their references to many other works, head direction information can directly impact CA1 neurons. Thus, the information provided is not novel. More importantly, it is incomplete to the point of being anecdotal. My personal opinion is that it lacks enough validity to be included as a refereed set of findings. The connection to the paper's discussion of the emergence of spatial tuning in PoSub is weak at best.

Finally, the authors need to revisit the scheme employed for the data of figure 2A. The relevant

contrasts are not well described in the text and the color-coding for different neuron types does not work well. It would be very difficult for a reader to get the point of the figure. Also, why is the uncorrected spatial information score used?

The authors have improved the manuscript in many respects. This can be a relatively important paper concerning spatial versus directional tuning in PoSub. As the emphasis in PoSub work has been placed mainly on head direction neurons, this paper could go some way to breaking new ground on the function of this region. However, I still have significant reservations concerning a key aspect of the work (HD tuning in PoSub), how it compares to that observed in prior work (where similar phenomena must be present), and the appropriateness of the data presented in figure 5. I recommend the authors take the opportunity to provide a more definitive account of the firing properties of their PoSub neurons and recommend that figure 5 and associated text be removed.

We thank the reviewer for his kind appreciation of our work. We have now decided to remove the last part of the story and to publish it separately. We have also made the requested changes to Figure 2 and included additional examples to demonstrate that PoSub neurons are primarily driven by a 'pure' HD signal.

In the revised manuscript, the authors provide some degree of information concerning the distribution of sensitivity of ADn and PoSub neurons to "spatial" information such as that provided by specific environmental borders. In particular, figure 2B shows that the border modulation scores for PoSub neurons fall on a continuum broader than that for the ADn neurons and with a higher overall mean. But where the PoSub distribution breaks away from the ADn distribution, the border modulation values are already near that for the PoSub neuron depicted in figure 1B. The problem is that I do not see how this neuron, nor the other two example neurons (figure 1E,F) from PoSub, can be clearly characterized as "head direction" neurons in the first place. There is extremely little firing within the central portions of the arena as well as along the west wall. What is there to indicate that the neuron is not responding to the presence of the east wall in close proximity to the left side of the animal? Is the head direction tuning curve not simply an epiphenomenon? I do understand that the authors are trying to say that the PoSub is an area where "spatial" signals, be they allocentric or egocentric, can strongly influence neurons that may be strongly influenced by head direction signals. But the authors should at least acknowledge in discussion that at some point, a neuron should not be recognized as a head direction neuron at all. It might help to include examples of PoSub neurons whose tuning is not strongly affected by border identities or proximities. Do PoSub "head direction" neurons with border modulation scores of less than 0.1 exhibit much firing in the center of the arena or do they simply have more similar firing patterns across parallel borders? If the latter is true, then it would seem that virtually none of the recorded PoSub neurons should be considered head direction neurons, but simply as neurons impacted by several spatial variables with some influence by head direction. The authors stated purpose in publishing this work is to show that PoSub "head direction" neurons are much more "spatial" in their firing patterns and so a more thorough consideration of the true influence of head direction is necessary.

We understand the concerns raised by the referee. We think part of the confusion stems from the strong emphasis we put on the PoSub neurons that were the most influenced by non-HD signals. However, many PoSub HD neurons are still 'pure' HD neurons, and the conjunction of HD and spatial correlates we report is continuously distributed. First, we have now included two examples of HD cells

recorded in the PoSub in Supplementary Fig 5 (A&B). These examples are also in line with the referee's suggestion **"to include examples of PoSub neurons whose tuning is not strongly affected by border identities or proximities"**. The cell displayed in panel Fig S5B is from the same session as the two cells of Figure 1A-D. It is not modulated by border, unlike the one displayed in Figure 1A and its spike train is mostly explained by its HD tuning curve only. Again, we show this example as a way to rule out issues with spatial sampling. It also shows that neuronal activity within the PoSub was continuously distributed from 'pure' HD cells, presumably under the direct influence of ADn, and cells that were classified as HD neurons based on their HD tuning curves but showed conjunctive body-wall relationship. For the sake of clarity, we have now added the following sentence to the result session:

However, not all HD neurons in the PoSub were modulated by border and egocentric information. Overall, the effect is continuously distributed from 'pure' HD cells (green dots in Figure 2B, left panel) to border-modulated HD neurons (red dots; see also Supplementary Figure 5).

However, we looked for cells that had clear 'place fields', not necessarily along the walls. Except a few not really convincing examples, we could not conclude that there were place-by-HD neurons in our recordings. Modulation by border (or, equivalently, egocentric) information remains the best explanation for the higher unbiased spatial information of HD neurons in the PoSub than in the ADn. This is also supported by the fact that the PoSub is one of the main output structures of the ADn. It is thus fair to assume that a large number of PoSub neurons is strongly driven by a 'pure' HD signal.

I still find the data of figure 5 to be an unnecessary distraction. As the authors show through their references to many other works, head direction information can directly impact CA1 neurons. Thus, the information provided is not novel. More importantly, it is incomplete to the point of being anecdotal. My personal opinion is that it lacks enough validity to be included as a refereed set of findings. The connection to the paper's discussion of the emergence of spatial tuning in PoSub is weak at best.

We have now decided to remove the hippocampal data from the manuscript.

Finally, the authors need to revisit the scheme employed for the data of figure 2A. The relevant contrasts are not well described in the text and the color-coding for different neuron types does not work well. It would be very difficult for a reader to get the point of the figure. Also, why is the uncorrected spatial information score used?

We agree that the figure was not clear enough. We have now removed all mention to non-pyramidal neurons. In addition, as suggested by the referee, we now use unbiased spatial information. This is much more informative as it clearly shows that the two quantities (HD and unbiased spatial info) are not correlated.

REVIEWERS' COMMENTS:

Reviewer #2 (Remarks to the Author):

The latest version of the manuscript is much improved. The story/comparison is much cleaner and the new supplemental figures provide a much needed perspective on the nature of the full dataset. I'm in support of publication.